# Optimal Progressive Pitch for OneWeb Constellation with Seamless Coverage

**DOI:** 10.3390/s22166302

**Published:** 2022-08-22

**Authors:** Cheng Zou, Haiwang Wang, Jiachao Chang, Fengwei Shao, Lin Shang, Guotong Li

**Affiliations:** 1Innovation Academy for Microsatellites of CAS, Shanghai 201203, China; 2University of Chinese Academy of Sciences, Beijing 100049, China; 3School of Information Science and Technology, ShanghaiTech University, Shanghai 201210, China; 4Shanghai Yuanxin Satellite Technology Co., Ltd., Shanghai 201600, China

**Keywords:** LEO, GSO, broadband satellite constellation, progressive pitch, seamless coverage, interference, spatial separation

## Abstract

Large-scale broadband low earth orbit (LEO) satellite systems have become a possibility due to decreased launch costs and rapidly evolving technology. Preventing huge LEO satellite constellations from interfering with the geostationary earth orbit (GSO) satellite system, progressive pitch is a technique to avoid interference with the GSO satellite system that allows the LEO satellite system to maintain a certain angle of separation from the GSO satellite system. Aside from interference avoidance, there is also a need to ensure seamless coverage of the LEO constellation and to optimize the overall transmission capacity of the LEO satellite as much as possible, making it extremely complex to design an effective progressive pitch plan. This paper models an inline interference event and seamless coverage and builds an optimization problem by maximizing transmission capacity. This paper reformulates the problem and designs a genetic algorithm to solve it. From the simulation results, the strategy can avoid harmful interference to the GSO satellite system and ensure the seamless coverage of the LEO constellation, and the satellite transmission capacity is also maximized.

## 1. Introduction

COVID-19 has profoundly changed the world. The realization of a global Internet connection has been accelerated due to the increasing need for remote work and online education. Although the terrestrial network is so advanced, half the people still do not have Internet access [1]. This connectivity gap could not be ignored while the LEO (Mega Low Earth Orbit) constellation has the potential to deliver broadband Internet access anywhere in the world, which is seen as an efficient solution for equal connectivity worldwide. The LEO constellations are changing the traditional areas such as industrial IoT, agriculture, e-health and energy [2]. The possibilities of the LEO constellation have driven its development, which are detailed below.

The LEO constellation offers ubiquitous connectivity and is designed to provide services to rural and remote areas [3,4,5] where the terrestrial network is weak, which has been seen as a method to mitigate the digital divide.The LEO constellation network could provide a reliable and resilient link immune to natural disaster [6] and warfare.In big cities, the LEO network can support the terrestrial network with fair equal connection. The telecom operators could expand their network with the LEO constellation to provide a fast and reliable connection.Compared to geostationary earth orbit (GSO) satellites, LEO satellites have much lower orbital altitudes and therefore shorter delays. Compared to terrestrial networks, signals are transmitted via inter-satellite links (ISL), which significantly reduce propagation delays, compared with fiber optics.

With advances in technology and lower launch costs [7], the cost of building a Mega LEO constellation will be lower. SpaceX, Oneweb, Kuiper and other companies are building their broadband LEO constellations at a rapid pace. Although the LEO satellite has the potential to play a significant role in the future network [8], there are challenges to be overcome, such as the limited orbital resources, co-frequency interference and the complexity of the resource allocation [9]. Co-frequency is an inevitable challenge as the different communication systems might share the same frequency with the LEO constellation, such as the terrestrial network and the GSO network. In addition to different communication systems and services, co-frequency interference between LEO constellations also needs to be addressed. The reasonable use of spectrum resources requires the efforts of all operators who have overlapping spectra.

With the launch of a large number of spacecraft, orbital and frequency resources become very congested, making efficient use of non-renewable resources critical for the long-term exploitation of space [10]. The International Telecommunication Union (ITU) has limited the equivalent power flux density (EPFD) of non-geostationary earth orbit (NGSO) satellite systems in Article 22 of the Radio Regulations [11] to avoid unwanted interference from NGSO satellite systems to GSO satellite systems. The mitigation of large-scale LEO constellation interference to GSO satellite systems is one of the major technical concerns according to [12]. Spatial isolation [13], power control [14,15,16], system characteristics database, beamforming and cognitive radio [17,18,19,20] are commonly used interference avoidance methods for satellite systems.

The spatial isolation method includes the design of isolation zones, GSO avoidance arcs and changing satellite pointing methods [21]. The power control method can also be effective in suppressing interference with other systems. In Ref. [22], the authors introduce an adaptive power control technique for uplink and downlink scenarios to reduce inline interference between GSO and O3b satellite systems. In Ref. [23], the database approach is reviewed for the coexistence of NGSO satellite systems and other GSO satellite systems depending on the operational characteristics of the system (e.g., frequency assignment, orbital position and antenna pattern). Zhang et al. in Refs. [24,25] also conducted a series of explorations of cognitive radio on the spectrum coexistence of the NGSO satellite system and GSO satellite system. A spectrum-sensing technique for the NGSO satellite system to access the GSO satellite systems’ spectrum was developed in Ref. [25], which was adopted to detect whether the spectrum resource is occupied by the GSO satellite system and then to identify the specific power level used in the system.

Large-scale LEO satellite constellations generally adopt the method of spatial isolation to avoid interference with GSO satellite systems. The O3b [26] satellite system adopts the equatorial circular orbit constellation in order to avoid interference with the GSO satellite system. The latitude of the O3b system earth station is not less than 10 degrees, so that the angle of the O3b and GSO satellites with respect to the earth station meets a certain threshold. The Leosat [27] satellite uses a GSO arc avoidance angle, with a minimum GSO arc avoidance angle of 7.5 degrees. The Starlink satellite uses advanced phased array technology to provide communication services to ground users in order to avoid the interference to the GSO system that mainly adopts interference avoidance techniques [28] based on isolation angle and beamforming to suppress the interference in the inline scenario. For the OneWeb system, a progressive pitch-based approach is proposed, where the OneWeb satellite shifts its coverage area to lower latitudes by changing its attitude as the latitude of the satellite decreases, thus maintaining an angular separation between the OneWeb [29] system and the GSO satellite system.

Seamless global coverage is a key benefit of large-scale LEO satellite constellations. Due to the large-scale constellation, the Starlink system can provide multiple coverage of the Earth’s surface. It is possible to achieve a blind spot-free coverage of the Earth despite the presence of exclusion zones. For OneWeb’s progressive pitch technology, it is easy to lead to communicate blind spots between adjacent satellites because the satellite coverage area is changed. Communication blind spot elimination also makes the design of the progressive pitch strategy more complicated. The interference avoiding approaches based on spatial isolation are used in both of them, but they are totally different from each other as a result of different beam patterns and antennas. Recently, a progressive pitch scheme based on the OneWeb system to avoid interference to the GSO satellite system in the downlink is given in the literature [30], but the scheme does not guarantee seamless coverage of LEO satellite systems. In this paper, the progressive pitch technique is redesigned based on the seamless coverage characteristics of the LEO satellite constellation.

One of the most important features of the broadband LEO satellite constellation is to provide Internet access for seamless global coverage. When considering interference avoidance for GSO satellite systems, it is necessary to ensure that no blind spots exist within its coverage. Although there is a t of literature on interference to GSO system from LEO constellations, there is still very little research on interference avoidance techniques that can guarantee seamless coverage. This article proposes an interference avoidance scheme that enables seamless coverage.

We analyze the downlink EPFD of the NGSO satellite system on the orbital plane and evaluate each satellite’s contribution to the EPFD. Based on the results of this analysis, the focus is on the EPFD in the inline scenario.Based on the spectrum coexistence scenario of the LEO constellation and GSO satellite system, the inline interference and seamless coverage are modeled. The average total communication capacity of the LEO system is maximized under the constraints of no harmful interference to the GSO system and no blind spots between adjacent satellites. Finally, the corresponding optimization problem is created.In order to obtain the optimal interference avoidance strategy, the problem is first discretized and reformulated. Then we solve the problem with a genetic algorithm that includes the design of the fitness function, encoding and operators.Based on the inline interference model and OneWeb system parameters, the off-axis angle thresholds are obtained without harmful interference. The variation of various metrics with satellite latitude is also given, which includes the total satellite communication capacity, the range of the coverage overlap between adjacent satellites and the minimum off-axis angle.

The rest of the paper is organized as follows. In Section 2, the inline interference and seamless coverage are modeled, and the corresponding optimization problem is formulated by maximizing the average total satellite communication capacity. In Section 3, the optimization problem is reformulated and solved by genetic algorithm. In Section 4, we evaluate the proposed algorithms by simulations. In Section 5, we conclude the study and provide an outlook on future research directions.

## 2. Problem Modeling

This section proposes a framework for an interference avoidance scheme for the LEO satellite to the GSO satellite system. First, it must ensure that the LEO satellite system does not generate unacceptable interference to the system. Then it is necessary to ensure seamless coverage of the Earth by the LEO satellite constellation and finally to design the interference avoidance scheme by maximizing the average communication capacity of the system. The detailed notations used in this paoper are sumarized in Table 1.

We need to avoid inline interference of a single satellite according to the EPFD simulation of satellites in the same orbital plane in Appendix A because the EPFD maximum is determined by the satellite that contributes the most to the EPFD.

### 2.1. Beam Layout

In this paper, we focus on the polar orbit constellation. The OneWeb satellite has *N* user beams where *N* equals 16. Each beam is denoted as $b_{i}$, i∈{1,2,⋯,N}, and its shape is highly elliptical. The coverage pattern is shown in Figure 1, where the major and minor axis of the elliptical beam are along the east–west and north–south directions, respectively. The coverage of Figure 1 is a simplified version of a spherical cap in Earth’s surface, and the flattened coverage in this figure is used to specify how the beams are deployed, such as the frequency plan, the layout of beams and their elevation angles. The beams of the same color in the satellite coverage have the same frequency. The set of frequencies is F, where $F=\{f_{1},f_{2},\cdot ,f_{m},\cdot ,f_{M}\}$ and *F* is 8 in the OneWeb system. *N* is an integer multiple of *M*, and the number of beams of the same frequency is $\frac{N}{M}$. It is assumed that the satellite is located in the northern hemisphere and moves southward. SSP represents the subsatellite point, where $BC_{i}$ denotes the center of the beam $b_{i}$, and CC denotes the center of the satellite coverage. The angle between SSP and CC as observed from the satellite is the pitch angle χ. For the $BC_{i}$, the elevation angle is $\chi +\left(2i-N-1\right)\mu_{b}$.

The antenna gain of the beam $b_{i}$ with respect to the point *E* is determined by the off-axis angles in the major and minor axis directions. To describe the antenna gain, off-axis angles $\mu_{i}$ in the minor axis and $\nu_{i}$ in the major axis are defined. The antenna gain of beam $b_{i}$ at point *E* is a function of $\mu_{i}$ and $\nu_{i}$, denoted as $G_{t}\left(\mu_{i},\nu_{i}\right)$.

The antenna model recommended by ITU-R S.1528 [31] is used to model the satellite radiation pattern for NGSO satellite operating in the fixed-satellite service below 30 GHz. According to ITU-R S.1528, the logarithm of $G_{n}\left(\Psi \right)$ is given by
(1)$$\left[G_{n}\left(\Psi \right)\right]=\left\{\begin{array}{cc} G_{max}-3{\left(\Psi /\Psi_{b}\right)}^{\alpha } & 0<\Psi \leq a\Psi_{b}\\ G_{max}+L_{N}+25\log\left(z\right) & a\Psi_{b}<\Psi \leq 0.5b\Psi_{b}\\ G_{max}+L_{N} & 0.5b\Psi_{b}<\Psi \leq b\Psi_{b}\\ X-25\log(\Psi ) & b\Psi_{b}<\Psi \leq Y\\ L_{F} & Y<\Psi \leq 90^{\circ }\\ L_{B} & 90^{\circ }<\Psi \leq 180^{\circ } \end{array}\right.,$$
where $X=G_{max}+L_{N}+25\log\left(b\Psi_{b}\right)$ and $Y=10^{0.04(G_{max}+L_{N}-L_{F})}$, $G_{max}$ is the maximum gain in the mainlobe (dBi) and $\Psi_{b}$ is the one-half the 3 dB beamwidth. The far-out side-lobe level, the back-lobe level and the main beam and near-in side-lobe mask cross point below peak gain are denoted as $L_{F}$, $L_{B}$ and $L_{N}$, separately. For $L_{N}=-25$, the values of *a*, *b* and α are $2.58\sqrt{1-0.6\log z}$, 6.32 and 1.5. The value of *z* is set to 1 to represent the gain of the minor axis of the transmitting antenna.

To describe the antenna envelope of an highly elliptical beam, the one-half 3 dB beamwidths in the minor and major axis of the antenna are denoted as $\mu_{b}$ and $\nu_{b}$, respectively. The elliptical antenna gain is simplified to the following
(2)$$G_{t}\left(\mu ,\nu \right)=\left\{\begin{array}{cc} G_{n}\left(\mu \right) & \nu \leq \nu_{b}\\ 0 & \nu >\nu_{b} \end{array}\right.,$$
where $G_{n}\left(\mu \right)$ denotes the antenna gain in the minor axis with the one-half beamwidth being $\mu_{b}$.

### 2.2. Interference Caculation

EPFD is a method adopted by the International Telecommunication Union Radiocommunication Sector (ITU-R) to evaluate radio frequency interference. To avoid unacceptable radio interference from NGSO systems to GSO systems, Article 22 of the ITU Radio Regulations places corresponding restrictions on EPFD from satellite system to a GSO satellite system. In accordance with the clause, the EPFD between the GSO satellite system and the NGSO satellite system is presented as
(3)$$epfd=10log_{10}\left(\sum_{i=1}^{N_{a}}10^{\frac{P_{i}}{10}}\times \frac{G_{t}\left(\Psi_{i}\right)}{4\pi d_{i}^{2}}\times \frac{G_{r}\left(\varphi_{i}\right)}{G_{r,max}}\right),$$
where $N_{a}$ denotes the number of transmitting stations of the NGSO satellite system, and $P_{i}$ is the input power of the antenna of the *i*th transmitting station. $G_{t}\left(\Psi_{i}\right)$ is the antenna gain of the *i*-th transmitting station of the NGSO satellite system with an off-axis angle of $\Psi_{i}$, and $d_{i}$ denotes the distance between the *i*th transmitting station of the NGSO satellite system and the victim receiving station. $G_{r}\left(\varphi_{i}\right)$ denotes the antenna gain of the receiving station with the off-axis angle of $\varphi_{i}$, and $G_{r,max}$ is the maximum antenna gain of it.

### 2.3. Inline Interference Model

The interference scenario is illustrated in Figure 2, where the GSO earth station, LEO satellite and GSO satellite are located on a straight line. The east–west boundaries of the coverage region of beam $b_{i}$ are marked by points A and B, respectively. The plane ABL is the cross section where the major axis of elliptical beam $b_{i}$ is located, and point C is the center of beam $b_{i}$. The elevation and azimuth angles are $\alpha_{i}$ and 0, respectively, where the elevation angle $\alpha_{i}$ is expressed as $\alpha_{i}=-\left(N-2i+1\right)\mu_{b}+\chi$.

The off-axis angles of point D with respect to the major-axis and minor-axis directions of beam $b_{i}$ are denoted as $\nu_{i}$ and $\mu_{i}$ respectively. The LEO satellite is located in the YOZ plane, and the longitude difference is σ between the GSO satellite and it. The EPFD of point D is shown as below
(4)$$p_{f}=\sum_{b_{i}\in S_{f}}\frac{p_{b}^{t}G_{t}\left(\mu_{i},\nu_{i}\right)}{4\pi d^{2}}\leq \frac{p_{b}^{t}G_{t}\left(\mu_{f,min},0\right)\left|S_{f}\right|}{4\pi h_{n}^{2}}.$$

Given the latitude ψ, longitude difference σ and pitch angle χ of the NGSO satellite, refer to Appendix B to calculate $\mu_{i}$ and $\nu_{i}$ of $b_{i}$. Suppose the central frequency of $b_{i}$ is *f*, $S_{f}$ is the set of beams with frequency *f*, and $\left|S_{f}\right|$ represents the number of beams with central frequency *f*, which equals $\frac{N}{M}$.

In the Inequality (Equation 4), $\mu_{f,min}$ is $min_{b_{j}\in S_{f},\left|\sigma \right|<\sigma_{max}}\mu_{j}$, and *d* represents the distance between the NGSO satellite and the GSO earth station. The input power of the reference bandwidth is denoted as $p_{t}^{b}$, which equals to $10^{EIRP-10log_{10}\left(\frac{B_{w}}{B_{ref}}\right)}$, where $B_{w}$ is the bandwidth, and $B_{ref}$ is the reference bandwidth regarding the caculation of EPFD. Then, we can have the inequality
(5)$$\begin{array}{cc} epfd_{f} & =10\log_{10}\left(p_{f}\frac{G_{r}\left(0\right)}{G_{r,\max}}\right)\leq EIRP-10\log_{10}\left(\frac{B_{w}}{B_{ref}}\right)+10\log_{10}\left(\left|S_{f}\right|\right)\\ & +10\log_{10}\left(\frac{G_{t}\left(\mu_{f,min},0\right)}{G_{t}\left(0,0\right)}\right)-10\log_{10}\left(4\pi h_{n}^{2}\right), \end{array}$$
where $G_{r,max}$ denotes the maximum antenna gain of the GSO earth station, and $G_{r,max}=G_{r}\left(0\right)$.

To make the downlink EPFD satisfy the corresponding limits, it is ensured that the right-hand side of (Equation 5) is smaller than $epfd_{th}$. Therefore, the off-axis angle in minor axis of the beams should be less than $\mu_{th}$. According to Equation (Equation 6), the off-axis angle threshold ($\mu_{th}$) can be obtained. In order to circumvent interference from LEO satellites to the GSO satellite system, the off-axis angle of the minor axis of all activated beams of LEO satellites should be less than the threshold.
(6)$$10\log_{10}\left(\frac{G_{n}\left(\mu_{th}\right)}{G_{n}\left(0\right)}\right)=epfd_{th}+10\log_{10}\left(4\pi h_{n}^{2}\frac{B_{w}}{B_{ref}}\right)-EIRP-10\log_{10}\left(\left|S_{f}\right|\right)$$

### 2.4. Seamless Coverage Model

This section mainly focuses on the seamless coverage provided by the LEO satellite constellation and, to simplify the analysis process, on the orbital plane coverage provided by the LEO satellite constellation. The coverage of the orbital plane of a single LEO satellite is shown in Figure 3. In order to achieve seamless coverage on the orbital plane, it is necessary to ensure that there are overlapping parts in the coverage area of two adjacent satellites, and the size of the overlap is affected by the latitude difference between the adjacent satellites and their interference avoidance strategy.

The coverage model of the LEO satellite is shown above. The arc AB is the coverage corresponding to all beams, and the arc AD corresponds to the off beams. *C* is the center of the arc AB, χ is the pitch angle, and the number of off beams is *K*. The order of shutting beams is from the high latitude to the low latitude, and the set of off beams can be represented by $\left\{b_{1},b_{2},\cdots ,b_{K}\right\}$. Both the pitch angle χ and the number of off beams *K* are functions of the satellite latitude ψ. The latitude of the upper and lower boundaries of the LEO satellite coverage area are $\psi_{up}$ and $\psi_{down}$, separately. The elevation angles of points *B* and *D* relative to the LEO satellite are functions of the latitude of the LEO satellite, and the elevation angle expressions of *D* and *B* are expressed as follows
(7)$$\alpha_{up}\left(\psi \right)=\chi \left(\psi \right)+\left(2K\left(\psi \right)-N\right)\mu_{b},$$
(8)$$\alpha_{down}\left(\psi \right)=\chi \left(\psi \right)+N\mu_{b}.$$

The expressions of $\psi_{up}$ and $\psi_{down}$ are shown as below
(9)$$\psi_{up}\left(\psi \right)=\psi -\arcsin\left(\frac{R_{e}+h}{R_{e}}sin\alpha_{up}\left(\psi \right)\right)sign\left(\alpha_{up}\left(\psi \right)\right),$$
(10)$$\psi_{down}\left(\psi \right)=\psi -\arcsin\left(\frac{R_{e}+h}{R_{e}}sin\alpha_{down}\left(\psi \right)\right)sign\left(\alpha_{down}\left(\psi \right)\right).$$

For any elevation angle α the LEO satellite, the corresponding latitude $\psi^{{}^{\prime }}$ expression is shown as
(11)$$\psi^{{}^{\prime }}=\psi -\arcsin\left(\frac{R_{e}+h}{R_{e}}sin\alpha \right)sign\left(\alpha \right).$$

To achieve seamless coverage in the orbital plane, it is necessary to ensure that there is coverage overlap between neighboring satellites. The constraint can be expressed as
(12)$$\psi_{up}\left(\psi \right)-\psi_{down}\left(\psi +\delta \right)\geq \epsilon ,$$
where ϵ is the size of overlap, and δ is the latitude difference between neighboring satellites.

### 2.5. Performance Evaluation

The system capacity is a common measure of a communication system. The average capacity of the LEO system in the orbital plane is proposed to simplify the analysis of the average communication capacity. For an active beam *i*, (i>K(ψ)), the elevation angle expressions for the upper and lower boundaries of the beam coverage are formulated as follows
(13)$$\alpha_{up}^{i}\left(\psi \right)=\chi \left(\psi \right)+\left(2i-2-N\right)\mu_{b},$$
(14)$$\alpha_{down}^{i}\left(\psi \right)=\chi \left(\psi \right)+\left(2i-N\right)\mu_{b}.$$

According to the expression (Equation 11) and Equations (Equation 13) and (Equation 14), the latitude corresponding to the upper and lower boundaries of $b_{i}$ can also be obtained, and the boundaries are denoted as $\psi_{up}^{i}$ and $\psi_{down}^{i}$ here.
(15)$$c_{i}\left(\psi \right)=B_{w}\int_{\psi_{down}^{i}}^{\psi_{up}^{i}}\log_{2}\left(1+\frac{p_{t}G_{n}\left(\alpha \left(x\right)\right)G_{r,max}{\left(\frac{c}{4\pi d(x)f}\right)}^{2}}{kTB+\sum_{b_{j}\in S^{b_{i}},b_{j}\neq b_{i}}p_{t}G_{n}\left(\right|\alpha_{j}-\alpha \left(x\right)\left|\right)G_{r,max}{\left(\frac{c}{4\pi d(x)f}\right)}^{2}}\right)dx$$

The average communication capacity of $b_{i}$ can be expressed as Equation (Equation 15), where $G_{r,max}$ is the maximum gain of receiving antennas. The noise of the communication system is kTB, where *k* is the Boltzmann constant, *T* is the system noise, and *B* beam bandwidth; d(x) is the distance from the point with latitude *x* in the coverage to the satellite, and α(x) is the elevation angle of the LEO satellite with respect to latitude *x*. $S^{b_{i}}$ is the set composed of beams of the same frequency as $b_{i}$, and *c* is the speed of radio wave in vacuum.

The total capacity of the LEO satellite is the sum of the average capacity of all activated beams, which is expressed as in Equation (Equation 16)
(16)$$c\left(\psi \right)=\sum_{i=K(\psi )}^{N}c_{i}\left(\psi \right).$$

To evaluate the performance of interference avoidance schemes, the integral of the total communication capacity of LEO satellite over latitude is used as an evaluation metric with the expression
(17)$$c\left(\psi \right)=\int_{0}^{90}\sum_{i=K(\psi )}^{N}c_{i}\left(\psi \right)d\psi .$$

### 2.6. Problem Formulation

According to the interference and coverage model, the average communication capacity of the system is maximized to define the optimization problem.

To circumvent the interference from the LEO system to the GSO system, it is necessary to make the off-axis angle of each beam in the inline scenario larger than the isolation angle threshold $\mu_{th}$. In order to ensure seamless coverage of the LEO satellite, a certain overlapping coverage area needs to be guaranteed between adjacent satellites.

Under the constraints of interference avoidance and seamless coverage, the maximized average communication capacity of the LEO satellite system can be expressed as the following optimization problem.
(18)P1:maximize{χ(ψ),K(ψ)}∫090∑i=K(ψ)Nci(ψ)dψ(18a)subjectto:0≤χ(ψ)≤χmax,(18b)0≤K(ψ)≤N,(18c)K(ψ)∈Z+,(18d)ψup(ψ)−ψdown(ψ+δ)≥ϵ,(18e)ψ+δ≤90,(18f)ψ≥0,(18g)μmini(ψ)≥μth,∀i∈{1,2,·,N}.

In the above optimization problem, the constraints include constraints on the maximum pitch angle (Equation 18a) and the maximum number of closed beams (Equation 18b), in addition to constraints on the seamless coverage (Equation 18d) and the harmful interference (Equation 18g). In (Equation 18g), $\mu_{min}^{i}\left(\psi \right)$ equals to $min_{\left|\sigma \right|<\sigma_{max}}\mu_{i}\left(\psi \right)$. For the progressive pitch scheme, both the LEO satellite pitch angle and the number of off beams increase as the satellite latitude decreases.

## 3. Algorithm Design

This section shows how the problem can be solved using a genetic algorithm. First, we give a brief introduction to the genetic algorithm, and then define the genetic algorithm fitness function and associated operators according to the problem.

### 3.1. Genetic Algorithm

Genetic algorithms are a class of heuristic algorithms based on evolutionary iteration of populations to obtain better solutions. In genetic algorithms, each individual is genetically determined by a specific way of encoding. The main operators in the evolutionary process of the algorithm include crossover, mutation and selection operators. Since genetic algorithms have been successful in many combinatorial optimization problems with NP-hard, the genetic algorithm has been used to solve the optimal interference avoidance strategy.

### 3.2. Problem Reformulation

According to the problem description in the previous section, the problem is a combinatorial optimization problem. It is very difficult to solve the original problem directly. To simplify solving the problem, we solve this optimization problem by a sampling of latitudes from 0 to 90 degrees discretely at equal intervals, and the phase difference between adjacent satellites is an integer multiple of the sampling interval. We denote the sampling interval as Δψ and δ=LΔψ, where *L* is an integer. The discrete optimization problem can be described as the following optimization problem.
(19)P2:maximize{χp,Kp}1⌊90Δψ⌋∑p=1⌊90Δψ⌋∑i=KpNcip(19a)subjectto:0≤χp≤χmax,(19b)χp≥χp+1,(19c)0≤Kp≤N,(19d)Kp≥Kp+1,(19e)Kp≤∈Z+,(19f)ψdownp−ψupp+L≥ϵ,(19g)(p+L)Δψ≤90,(19h)μminip≥μth,∀i∈{1,2,·,N},

By discretizing the latitude, we transform the optimization problem P1 into the optimization problem P2 in which $\left[p\right]$ means (pΔψ). Since χ(ψ) and K(ψ) increase with decreasing ψ, the constraints (Equation 19a) and (Equation 19c) need to be satisfied.

### 3.3. Fitness Function

Genetic algorithms determine the selection and reproduction of populations based on the fitness function. Defining the fitness function of a genetic algorithm requires considering the objective function of the optimization problem and whether the constraints are satisfied. For this optimization problem P2, the fitness function f(χ,K) is defined as follows
(20)$$f\left(\chi ,K\right)=\frac{1}{\left\lfloor \frac{90}{\Delta \psi }\right\rfloor }\sum_{p=1}^{\left\lfloor \frac{90}{\Delta \psi }\right\rfloor }\sum_{i=1}^{N}c_{i}\left[p\right]+E\sum_{p=1}^{\left\lfloor \frac{90}{\Delta \psi }\right\rfloor }I\left(\left(\chi \left[p\right],K\left[p\right]\right)\in C_{p}\right),$$
where $C_{p}$ represents the constraint of problem P2, and *E* represents the cost factor of constraint violation. *I* is an indicative function. When $\left(I\left[p\right],K\left[p\right]\right)\in C_{p}$, *I* is 1, otherwise *I* is 0, where the constraints $C_{p}$ include the constraints (Equation 19a), (Equation 19c), (Equation 19e), (Equation 19f), (Equation 19g) and (Equation 19h).

### 3.4. Encoding

The mapping from the problem solution to the genes is called encoding, and the form of encoding directly affects the performance of the algorithm. When solving problem P2 using a genetic algorithm, the optimization variables need to be encoded at each latitude, where the pitch angle is a continuous variable and the number of off beams is a discrete variable. A proper encoding method has been employed to represent progressive pitch scheme, and the difference of the optimization variables are encoded.

The difference of pitch angle between adjacent latitudes is coded, which can be expressed as $\chi \left[p\right]-\chi \left[p+1\right]$, and the adjustment of pitch angle is divided into fast adjustment and slow adjustment. The change of pitch angle for fast adjustment is TΔχ and for slow adjustment is Δχ, where *T* is an integer. The change of the number of off beams $K\left[p\right]-K\left[p+1\right]$ between adjacent latitudes is assumed to be at most 1. The whole action space at each latitude is denoted as *A*, where $A=\{a_{1},a_{2},a_{3},a_{4}\}$, and they are no adjustment, slow adjustment, fast adjustment and closing beam, respectively. The length of chromosome is $\left\lfloor \frac{90}{\Delta \psi }\right\rfloor$, and each gene on it has four expressions. We denote the chromosome as *S*, where $S=\{s_{1},s_{2},\cdots ,s_{l},\cdots ,s_{\left\lfloor \frac{90}{\Delta \psi }\right\rfloor }\}$ and $s_{l}\in A$, and the gene $s_{l}$ denotes the action of the latitude of 90−lΔψ. Therefore the pitch angle and the number of off beams for the latitude 90−lΔψ could be represented as the following,
(21)$$\psi_{90-l\Delta \psi }=\sum_{k\in \left\{1,2,\cdots ,l\right\},s_{k}\in \left\{a_{1},a_{2},a_{3}\right\}}s_{k},$$
(22)$$K_{90-l\Delta \psi }=\sum_{k\in \left\{1,2,\cdots ,l\right\},s_{k}\in \left\{a_{4}\right\}}s_{k}.$$

In order to obtain more precise adjustment of the progressive pitch scheme, the action space should be extended. The chromosome could be divided into two distinct parts: the mapping of the difference of pitch angle and the mapping of the difference of the number of off beams. However, this extension could impose burden on the searching of optimum progressive pitch scheme.

### 3.5. Genetic Operator Design

The chromosomes are guided by genetic operators to find an optimal solution for the problem. The genetic operators has been customized regarding the problem, consisting of crossover, mutation and selection, which are combined to find the perfect solution for the problem. The chromosomes are modified by operators toward a better solution in every iteration.

#### 3.5.1. Crossover

The crossover operation combines two chromosomes to generate new superior offspring, and the superior characteristics of the parent chromosome are inherited to the offspring by exchanging several genes. The exchange of genes could be chosen randomly from the entire chromosome sequence. The new offspring at iteration *t*, $C^{t}$, are generated from two parent chromosomes, $S_{i}^{t}$ and $S_{j}^{t}$. The specific process could be formulated as below: First of all, two parent chromosomes are chosen. Then the genes are randomly chosen to exchange. Finally the offspring is created as the parent chromosomes after exchange. The *l*-th element in the new offspring $C^{t}\left(l\right)$ is randomly chosen between $S_{i}^{t}\left(l\right)$ and $S_{j}^{t}\left(l\right)$, where $S_{i}^{t}\left(l\right)$ and $S_{j}^{t}\left(l\right)$ are drawn from the population of chromosomes. The crossover operation could be represented as the following,
(23)$$C^{t}\left(l\right)=\left\{\begin{array}{cccc} S_{i}^{t}\left(l\right) & rand()\leq CR & or & l=randn(L)\\ S_{j}^{t}\left(l\right) & rand()>CR & and & l\neq randn(L) \end{array}\right.,$$
where CR is crossover rate and rand() is random value drawn from [0,1] uniformly. The larger CR is, the more $S_{i}^{t}\left(l\right)$ contributes, otherwise $S_{j}^{t}\left(l\right)$ contributes more. The length of the chromosome is denoted as *L* and equals $\left\lfloor \frac{90}{\Delta \psi }\right\rfloor$, and randn(L) is to draw an integer from [1,L]. Therefore at least one element is from the chromosome $S_{i}^{t}\left(l\right)$, which avoids the ineffective crossover.

#### 3.5.2. Mutation

In order to prevent the genetic algorithm from falling into local optimal solutions during the optimization process, it is necessary to mutate the individuals during the search process. Regarding the *i*-th chromosome $S_{i}^{t}$, the mutation operation is shown as the following: firstly an integer *l* is draw from [1,L], then $S_{i}^{t}\left(l\right)$ is replaced with a new gene chosen randomly from action space *A* excluding $S_{i}^{t}\left(l\right)$. Finally, the chromosome after mutation could be obtained, with one gene being different from the original one.

#### 3.5.3. Selection

The selection operation selects good chromosomes from the old population in a certain way to form a new population in order to reproduce and get the next generation of the population. In this paper, we select the top twenty percent of individuals from the population in terms of fitness and then sample the selected individuals to obtain a new population. Therefore, the new population is generally better than the parent population regarding fitness.

### 3.6. Complexity Analysis

The genetic algorithms are adopted frequently, but there are no general analysis for all genetic algorithms. The complexity of genetic algorithms include the number of iterations and the cost of each iteration. For each iteration, crossover, mutation and selection operations are performed. Assume the size of the population is *M*. The complexity of the above operators are O(M), O(M) and $O(M^{2})$, respectively; therefore the overall complexity is equivalent to $O(M^{2})$. It is difficult for the number of iterations to reach convergence and complex to find the complexity of iteration. Some interesting findings are in shown in [32,33]. Since the complexity is hard to obtain, in many cases, it is demonstrated by simulations.

## 4. Numerical Simulation and Results

This section evaluates the results and the performance of the algorithm, including a description of the interference scheme based on a genetic algorithm and related parameters. Then the convergence of the genetic algorithm applied to the progressive pitch strategy is analyzed. The algorithm is evaluated based on the total communication capacity, angular isolation and overlap coverage metrics of the satellite system. Finally, the algorithm is compared with other interference avoidance methods, which can have an effective guarantee of communication capacity of satellites under the condition of seamless coverage and no harmful interference to GSO satellite systems.

### 4.1. The Simulation Scenario

In this section, the OneWeb system model is used as input to design the interference avoidance scheme based on the genetic algorithm. The OneWeb system is used as a reference system. We have given a detailed description of the parameters of the system. In addition, the settings of the relevant parameters of the genetic algorithm are also given here. The specific parameters are summarized in Table 2.

To avoid the unacceptable interference with the GSO satellite system, the off-axis angle threshold needs to be determined. Based on the beamwidth and frequency of the OneWeb system, we are able to obtain the antenna gain of the OneWeb satellite beam in the minor axis. Then according to expression (Equation 6), we can obtain the off-axis angle threshold μth, which corresponds to an antenna gain that is 27.1 dB lower relative to the maximum antenna gain. The relative antenna gain attenuation of 27.1 dB corresponds to an off-axis angle of 11.5°, which has been given in Table 2.

The number of satellites in each orbital plane *N* highly affects the seamless coverage of the LEO constellation. The seamless coverage is ensured by the overlap between adjacent satellites. The relationship between coverage overlap and the number of satellites has been shown in Table 3, without considering the harmful interference into the GSO satellite system. The overlap denotes the geocentric angle of the shared coverage between adjacent satellites in the same orbital plane, which increases with the number of satellites.

In the above table, the seamless coverage in the orbital plane could be ensured. While the co-frequency interference could not be ignored, it is very important to ensure the seamless coverage under progressive pitch. In order to ensure there exists a progressive pitch scheme enabling the global coverage, we study the progressive pitch under 48 satellites per orbit plane.

### 4.2. The Convergence of the Genetic Algorithm

Figure 4 shows the relationship between the number of iterations and the fitness. It can be seen that the fitness of the best individual reaches convergence after 50 iterations. According to the chromosome of the optimal individual of the population after reaching convergence, we can obtain the interference avoidance strategy of the optimal individual by decoding this chromosome, and the curve of the pitch angle and the number of closed beams of this interference avoidance strategy is shown in Figure 5a,b.

In order to demonstrate the effectiveness, several progressive pitch schemes have been compared. A brief introduction of the proposed schemes is as the follows:Interference-only: An interference avoidance strategy that considers only interference constraints as given in the literature [30] ignores the seamless coverage.Brutal force: This method could achieve the seamless coverage with the interference constraints. Two parameters of latitude need to be determined: one parameter identifies the latitude where the LEO satellites start to tilt their attitude, the another is the latitude that the pitch angle starts to be same with the interference-only scheme.GA: The genetic algorithm scheme with random initialisation could ensure the seamless coverage without generating harmful interference to the GSO satellite system, but it has no advantages compared to the brutal force scheme. The action space for *A* is {0,+0.5,+1.5,+1}, of which the previous three elements represent the changes of pitch angle, and the last element represents the changes of the number of off beams.GA + brutal force: This method fine tunes the brutal force scheme with a genetic algorithm to further extend the capacity of the LEO satellite. The initial scheme of GA + brutal force is the same with the brutal force scheme. The action space for *A* is {0,+0.2,−0.2,+0.6,−0.6,+1,−1}, of which the previous five elements represent the changes of pitch angle and the last two elements represent the changes of the number of off beams.

Figure 5a,b show the variations of the satellite pitch angle and the number of off beams along the latitude in the northern hemisphere, and the interference avoidance strategy in the southern hemisphere is symmetric with the northern hemispere. The maximum values of the number of off beams are the same for the above methods, while the maximum value of the pitch angle in the approaches based on a genetic algorithm is smaller than the approaches of brutal force and interference-only. As LEO satellites orbit to lower latitudes, both pitch angle and the number of off beams increase monotonically, except for the GA + brutal force approach. Combining with Figure 5a,b, the pitch angle of the LEO satellite in the interference-only progressive pitch approach is smaller than other approaches before the satellites turn off the beams. In the interference-only approach, the seamless coverage constraint could not be ensured, while in other methods the seamless coverage is satisfied. In order to enable the seamless coverage, the coverage of LEO satellite needs to be shifted in advance toward the equator to provide more overlap between the adjacent satellite in a lower latitude.

### 4.3. The Results of the Proposed Algorithm and Comparison

This section compares the above interference avoidance strategies in terms of different metrics, such as the total satellite communication capacity, the minimum off-axis angle of the LEO satellite beams, and the overlapping coverage between the adjacent satellites in the same orbit plane. In order to verify whether the interference avoidance strategy satisfies the corresponding constraints and to evaluate the total communication capacity of the LEO satellite, we performed the related simulations regarding different progressive schemes and the results shown in the Figure 6a–c.

In Figure 6a, the satellite communication capacities have the same trend in the above progressive pitch schemes. The communication capacities decrease slowly at the high latitude; however, when the satellite approaches the lower latitude, the communication capacity decreases significantly, mainly caused by turning off the beams. The shutdown of the satellite beam occurs when the pitch angle reaches its maximum in the interference-only and brutal force schemes. Because of the high dimensional action space, it is difficult to find a good enough solution without an excellent initial guess. While in the GA + brutal force scheme, we take the brutal force scheme as the initial guess and use the genetic algorithm to optimize the solution. Since the capacity of the brutal force scheme nearly reaches the limit of the interference-only scheme, the improvement of the optimized solution by the genetic algorithm is very limited.

Figure 6b,c verify whether the above strategies satisfy the conditions of interference avoidance and seamless coverage, respectively. According to the results in Figure 6b, the minimum off-axis angle of the beams in the minor axis is greater than the angular threshold for all the approaches. There is a natural spatial separation between the LEO and GSO satellite systems at higher latitudes, and the off-axis angles of the LEO satellite beams are larger. The small bumps at low latitudes are mainly caused by turning off the satellite beams. However, in Figure 6c, interference avoidance-only progressive pitch strategy does not satisfy the constraint of seamless coverage. This also shows that designing an interference avoidance strategy by considering only interference avoidance is likely to result in a blind spot in satellite services. When designing a satellite interference avoidance strategy, seamless coverage consideration should not be neglected.

Based on the above simulation results, the interference avoidance scheme proposed in the paper can avoid interference to the GSO satellite system while ensuring seamless coverage; and the capacity of the proposed schemes nearly equals the capacity of interference-only progressive pitch scheme.

## 5. Conclusions

This paper mainly analyzes the downlink interference from the OneWeb system with a highly elliptical beam to the GSO satellite system and proposes an interference analysis method. From the distribution of EPFD in the orbital plane, a single satellite inline interference model is proposed. The corresponding off-axis angle threshold (μth) can be obtained, which is the basis for designing the spatial isolation method. In addition to satisfying the angular isolation of Oneweb satellites, it is important to ensure seamless coverage of the constellation and to use genetic algorithms to maximize the average total satellite communication capacity. The design of the encoding and operator of the genetic algorithm effectively simplifies the solution of this problem.

According to the simulation results, the progressive pitch strategy based on a genetic algorithm can effectively maximize the communication capacity of the LEO satellite under the conditions of interference avoidance and seamless coverage. The performance of the proposed genetic algorithm with the initial guess of brutal force scheme could approach the limit of the interference-only scheme.

This paper presents an interference avoidance strategy designed to ensure seamless coverage of the LEO satellite constellation. However, the interference avoidance only includes changing the satellite pointing and turning off beams and does not study the interference avoidance strategy based on power control, which is a very effective way for interference avoidance. In future research, we will focus on interference avoidance strategies with power control, which will bring benefits to the system performance improvement.

## Figures and Tables

**Figure 1 sensors-22-06302-f001:**
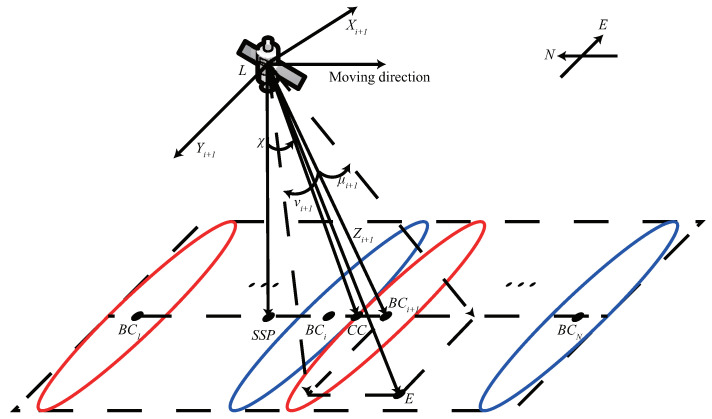
The coverage of LEO satellite with the pitch angle being χ.

**Figure 2 sensors-22-06302-f002:**
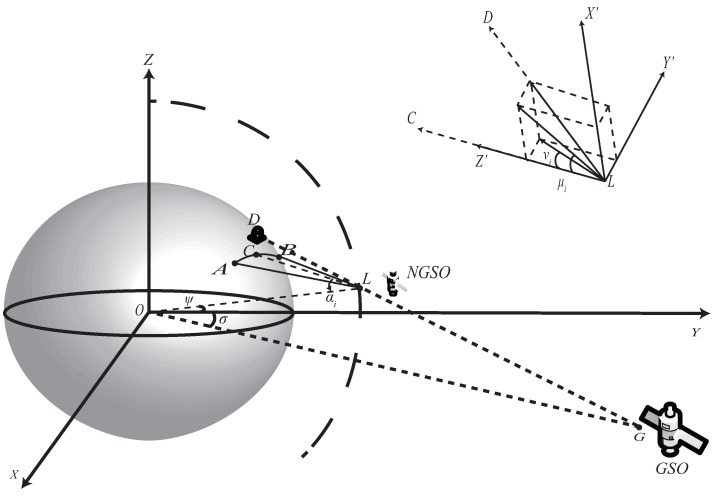
Downlink inline interference from NGSO satellite to GSO earth station.

**Figure 3 sensors-22-06302-f003:**
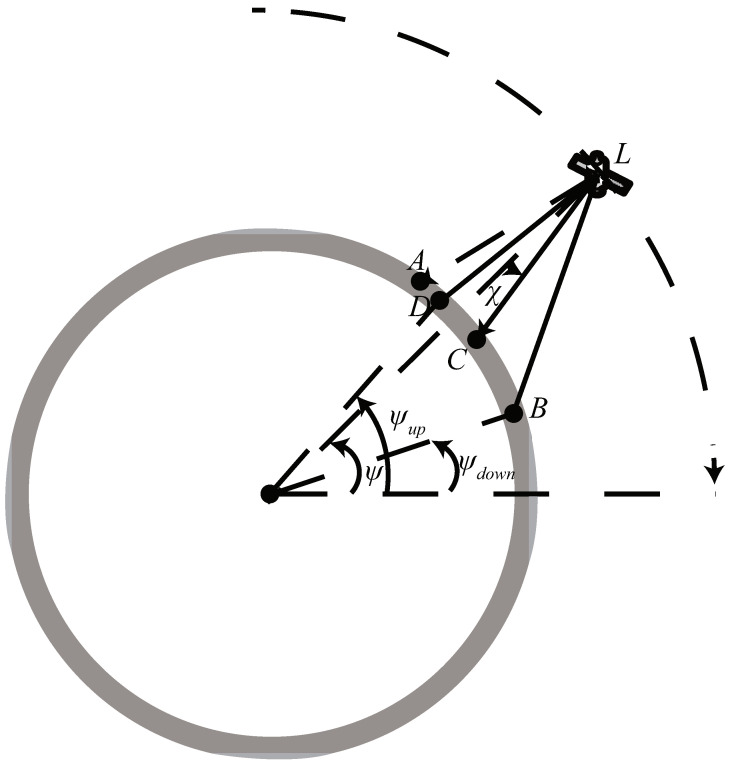
The coverage of LEO satellite in orbit plane.

**Figure 4 sensors-22-06302-f004:**
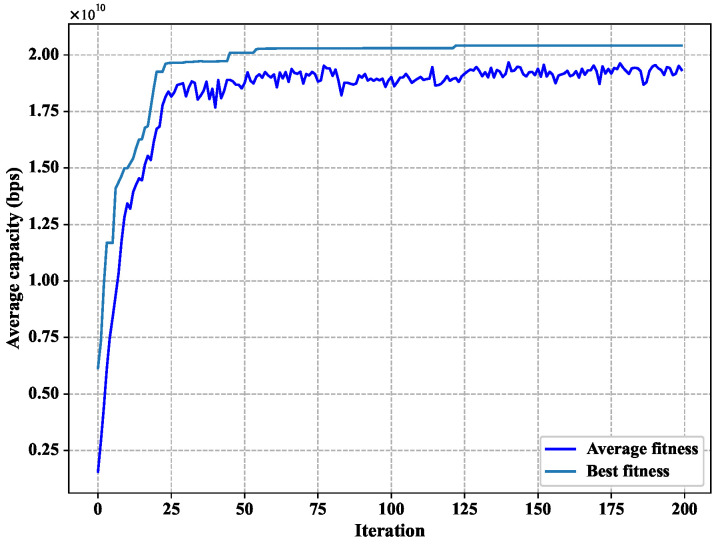
Convergence of the genetic algorithm.

**Figure 5 sensors-22-06302-f005:**
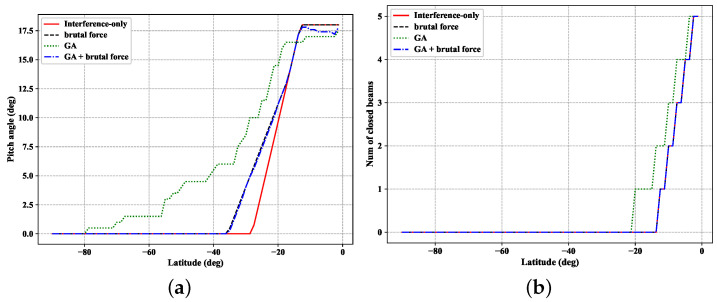
The scheme of progressive pitch: (**a**) pitch angle vs. the latitude of LEO satellite; (**b**) number of off beams vs. the latitude of the LEO satellite.

**Figure 6 sensors-22-06302-f006:**
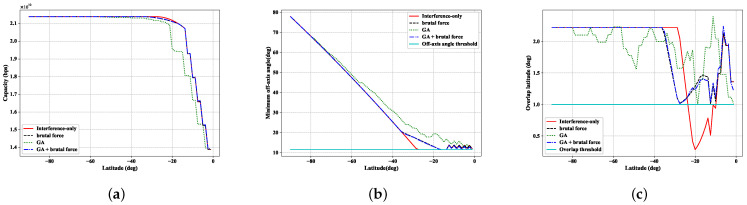
The comparisons of different schemes in terms of total capacity, minimum off-axis angle and coverage overlap: (**a**) total capacity of all active beams of the LEO satellite; (**b**) minimum off-axis angle in the minor axis of LEO satellite beams; (**c**) coverage overlap between adjacent LEO satellites.

**Table 1 sensors-22-06302-t001:** Notations used in this paper.

Notation	Description
χ	The pitch angle of the LEO satellite.
$\alpha_{i}$	Elevation angle corresponding to the boresight of the beam $b_{i}$.
$\mu_{i}$	The off-axis angle in the minor-axis of the beam $b_{i}$.
$\nu_{i}$	The off-axis angle in the major-axis of the beam $b_{i}$.
$\mu_{b}$	The one-half 3 dB beamwidth in the minor axis of the highly elliptical beam
$\nu_{b}$	The one-half 3 dB beamwidth in the major axis of the highly elliptical beam
$G_{t}\left(\mu ,\nu \right)$	The antenna gain of LEO satellite transmitting antenna.
$p_{b}^{t}$	Input power of the LEO satellite transmiting antenna.
$S_{f}$	Beam set with frequency being *f*.
$G_{r}\left(\varphi \right)$	Antenna gain of the GSO earth station with the off-axis angle φ.
$\psi_{es}$	Latitude of GSO earth station.
σ	Longitude difference between GSO satelite and NGSO satellite.
*K*	The number of closed beams.

**Table 2 sensors-22-06302-t002:** Parameters of genetic algorithm.

Parameters	Values
LEO satellite orbit height (Km)	1200
GSO orbit height (Km)	36,000
The radius of earth (Km)	6371
Minimum downlink frequency (GHz)	10.7
Maximum downlink frequency (GHz)	12.7
Number of beams	16
Bandwidth (MHz)	250
EIRP (dBW)	34.6
Beamwidth in minor-axis (deg)	2.98
Beamwidth in major-axis (deg)	47.6
Downlink EPFD limit (dBW/m^2^ 40 KHz)	−160
Generations	200
Population size	100
Length of gene	72
Mutation rate	0.02
Crossover rate	0.5
Election rate	0.2
Off-axis angle threshold (deg)	11.5
Coverage overlap threshold (deg)	1
Max pitch angle (deg)	18
Slow adjustment value of pitch angle Δχ (deg)	0.5
Fast adjustment value of pitch angle LΔχ (deg)	1.5
Sampling interval in latitude Δψ (deg)	1.25

**Table 3 sensors-22-06302-t003:** Coverage overap between adjacent satellites in the same orbit.

Item	Case 1	Case 2	Case 3	Case 4	Case 5
*N*	40	42	44	46	48
overlap (deg)	0.72	1.15	1.54	1.89	2.22

## Data Availability

The data presented in this study are available on request from the corresponding author.

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
