# Peer review of "Optimal Progressive Pitch for OneWeb Constellation with Seamless Coverage"

_sensors, 2022, doi:10.3390/s22166302_

Round 1

Reviewer 1 Report

please see the upload file about the peer-review for authors.

Reviewer 2 Report

The paper entitled "Optimal Progressive Pitch for OneWeb Constellation with Seamless Coverage" covers the interesting resource allocation optimization problem related to LEO constellations (OneWeb in particular) with respect to existing GEO satellite services.

A few Major issues should be addressed:

1. Limiting the discussion to 40 satellites (in a sun-synch orbit of 1200km) reduces the problem significantly and by no means is close to global coverage. The simulation should consider realistic scenarios (not just sun-synch orbits) of at least a few thousand satellites.

2. A comprehensive compassion with Starlink is needed (the remark regarding the different approach of Starlink is not enough)

3. The genetic algorithm should be explained in more detail - in particular the cross-over function. 

4. It is very hard to evaluate the quality of the GA performance, comparing the results to some greedy heuristics would be beneficial.

5. The complexity state of the suggested optimization problem should be stated. Or at least a clear reference that covers the theoretical complexity state is required.

6. Sharing the simulation data - as a benchmark - might help others to evaluate the performance of the suggested GA. 

Reviewer 3 Report

Major comments:

1. The eleptical beams are not flat over the Earth's surface. The beam is large enough to have a spherical cap surface. I recommend the authors indicate that this is a simplification, which changes many of the equations derived. 

2. The motivation behin dusing LEO satellites might be trivial for researchers working in the field, however it is a relatively new topic to many readers. I suggest the authors detail the opportunities of LEO satellites. The authors may refer to the following citations:

https://ieeexplore.ieee.org/document/9755278

https://ieeexplore.ieee.org/document/1522108

https://ieeexplore.ieee.org/document/9502642

3. Figures are not detailed and require more information in terms of what they represent and the conclusions drawn from them

Minor comments:

1. The overall language of the paper is alright however the technical language requires more review especially in the system model to enhance readability.

Round 2

Reviewer 2 Report

The authors have addressed all the comments.

Reviewer 3 Report

Authors have addressed my comments